

# Probabilistic adaptation in changing microbial environments

Yarden Katz[1,2] and Michael Springer[1]

[1] Department of Systems Biology, Harvard Medical School, Boston, MA, United States
[2] Berkman Klein Center for Internet & Society, Harvard University, Cambridge, MA, United States

## ABSTRACT

Microbes growing in animal host environments face fluctuations that have elements of both randomness and predictability. In the mammalian gut, fluctuations in nutrient levels and other physiological parameters are structured by the host's behavior, diet, health and microbiota composition. Microbial cells that can anticipate environmental fluctuations by exploiting this structure would likely gain a fitness advantage (by adapting their internal state in advance). We propose that the problem of adaptive growth in structured changing environments, such as the gut, can be viewed as probabilistic inference. We analyze environments that are "meta-changing": where there are changes in the way the environment fluctuates, governed by a mechanism unobservable to cells. We develop a dynamic Bayesian model of these environments and show that a real-time inference algorithm (particle filtering) for this model can be used as a microbial growth strategy implementable in molecular circuits. The growth strategy suggested by our model outperforms heuristic strategies, and points to a class of algorithms that could support real-time probabilistic inference in natural or synthetic cellular circuits.

## INTRODUCTION

Outside the laboratory, microbes are faced with rich and changing environments. To improve their chances of survival, single microbial cells must adapt to fluctuations in nutrients and other environmental conditions. The mammalian gut, home to prokaryotic and eukaryotic microbes (*Walter & Ley, 2011*; *Parfrey, Walters & Knight, 2011*), is a striking example of a changing environment with elements of both randomness and order. Nutrients and metabolites may fluctuate stochastically in the gut, but these changes are structured by the host's physiology, diet and behavior (*Thaiss et al., 2014*; *David et al., 2014a*; *David et al., 2014b*). Cells that exploit this noisy structure and anticipate changes in their environment would likely gain a fitness advantage.

It remains unclear what the information processing capabilities of microbial populations are in such changing environments. To what extent are cells able to learn from their environment's history and use this information to predict future changes? How sophisticated are the resulting computations, and in what environments will they

Corresponding author
Yarden Katz,
yarden@hms.harvard.edu

lead to increased fitness? Insight into these questions would shed light on the type of environments cells were selected for and may guide the search for molecular mechanisms that implement adaptive computation. This direction could also inform how microbes become pathogenic. The yeast *C. albicans*, for example, can turn from a harmless human gut commensal to a pathogen depending on the host environment and its nutrient composition (*Brown et al., 2014*; *Kumamoto & Vinces, 2005*; *Cottier & Muhlschlegel, 2009*). A better understanding of how microbes like *C. albicans* perceive and adapt to their environment may suggest ways of manipulating the environment to control pathogenic growth.

Progress on these questions requires analysis at multiple levels of abstraction, as outlined by *Marr (1982)* for information-processing in the nervous system. First, the computational task solved by cells has to specified. For microbial adaptation, this would mean characterizing the space of possible changing environments and identifying the cellular strategies that would result in optimal growth in each environment. Second, the algorithms and representations that cells need to execute the growth strategy would have to be described. Finally, at the implementation level, we have to give an account of how molecular interactions give rise to the algorithm and the necessary representations. A complete account of microbial adaptation would ultimately integrate the three levels.

There has been much work on understanding the molecular and genetic determinants of microbial growth in changing environments (e.g., using experimental evolution *Poelwijk, De Vos & Tans, 2011*; *New et al., 2014*), but less on defining the abstract computational problem that microbes face when adapting to such environments. In this work, we focus on the computational and algorithmic aspects of adaptive growth in changing environments. We computationally characterize a set of structured dynamic environments, where fluctuations are driven by an unobservable mechanism ("meta-changing" environments), and derive an adaptive strategy for optimal growth in these environments. Our focus is on changing nutrient environments, since nutrient metabolism can serve as a model for microbial information-processing more broadly.

## Nutrient metabolism as a system for studying microbial information-processing

A natural context in which to study the microbial response to changing environments is metabolic adaptation to nutrients. Because of its strong effect on growth, the way cells adapt to nutrients is a highly selectable trait, either genetically in long-term changing environments (as shown by experimental evolution studies *Mitchell et al., 2009*; *Tagkopoulos, Liu & Tavazoie, 2008*) or epigenetically in environments that change on shorter time scales (*Stockwell, Landry & Rifkin, 2015*; *Jarosz et al., 2014*).

While the control of nutrient and carbon source metabolism has been studied extensively in yeast and other microbes (*Broach, 2012*), there is generally no simple mapping between the environment's nutrient composition and microbial cell state (such as the choice of which metabolic pathway to upregulate, or the rate at which to grow). The elaborate molecular machinery for nutrient sensing and uptake suggests that the mapping may be quite complex.

Some of the complexity arises from the fact that microbes prefer to consume some nutrients over others, and that distinct nutrients require different and sometimes mutually exclusive pathways to be expressed. Glucose is generally the preferred sugar and its presence inhibits the expression of pathways required to metabolize alternative sugars like galactose (*Gancedo, 1992*). In yeast, distinct glucose transporters are upregulated depending on glucose levels in the environment, which are sensed by dedicated glucose sensors Snf3 and Rgt2 (*Santangelo, 2006*; *Busti et al., 2010*; *Ozcan, Dover & Johnston, 1998*). Additionally, many promoters in diverse yeast species were shown to be responsive to the presence of different carbon sources in the environment (*Weinhandl et al., 2014*). It has also recently been shown that in environments containing multiple nutrients, cells might be sensitive to complex functions of nutrient levels. Yeast cells decide to turn on the machinery necessary to metabolize galactose (GAL pathway) based on the ratio of glucose to galactose levels in their environment (*Escalante-Chong et al., 2015*).

In addition to molecular complexity of nutrient signaling, there are also memory effects at play in the nutrient response. For example, prior exposure to galactose in yeast alters the rate at which the GAL pathway will be induced upon subsequent galactose exposures (*Stockwell, Landry & Rifkin, 2015*), and a similar memory phenotype has also been observed for lactose in glucose-lactose switching environments in bacteria (*Lambert & Kussell, 2014*). (It has also been suggested that some bacteria retain memory of their environment's history more broadly, both on short and long timescales *Wolf et al., 2008*.) The environment's nutrient history can also affect single-cell variation in gene expression. Biggar and Crabtree showed that depending on whether previously grown in glucose or raffinose, populations can exhibit single-cell variation in GAL pathway levels when switched to an environment containing a mixture of glucose and galactose (*Biggar & Crabtree, 2001*). Other lines of theoretical and experimental work showed that single-cell variation can be a form of "bet-hedging" that leads to increased fitness under certain conditions (*Jablonka et al., 1995*; *Veening, Smits & Kuipers, 2008*).

Taken together, the intricate molecular machinery underlying nutrient signaling, the effects of nutrient memory, and single-cell variability in response to fluctuations suggest that microbes process information about their environment (*Bowsher & Swain, 2014*), and take into account both the environment's history and their internal cell state in making a decision.

## Changing discrete environments and inference-based adaptation

To ask how the environment's history influences microbial decision-making, a number of theoretical and experimental studies have used changing discrete environments (*Jablonka et al., 1995*; *Lambert & Kussell, 2014*; *Stockwell, Landry & Rifkin, 2015*). A discrete environment is shown in Fig. 1A (top), where there are two alternating nutrients. Although natural environments are far more complex, discrete switches have been experimentally useful for uncovering mechanisms of nutrient memory (*Lambert & Kussell, 2014*; *Stockwell, Landry & Rifkin, 2015*). Also, in interacting with a host environment, microbes may sense some fluctuations as effectively discrete. For example, en route from the external environment to the gastrointestinal tract, microbes experience sharp shifts in

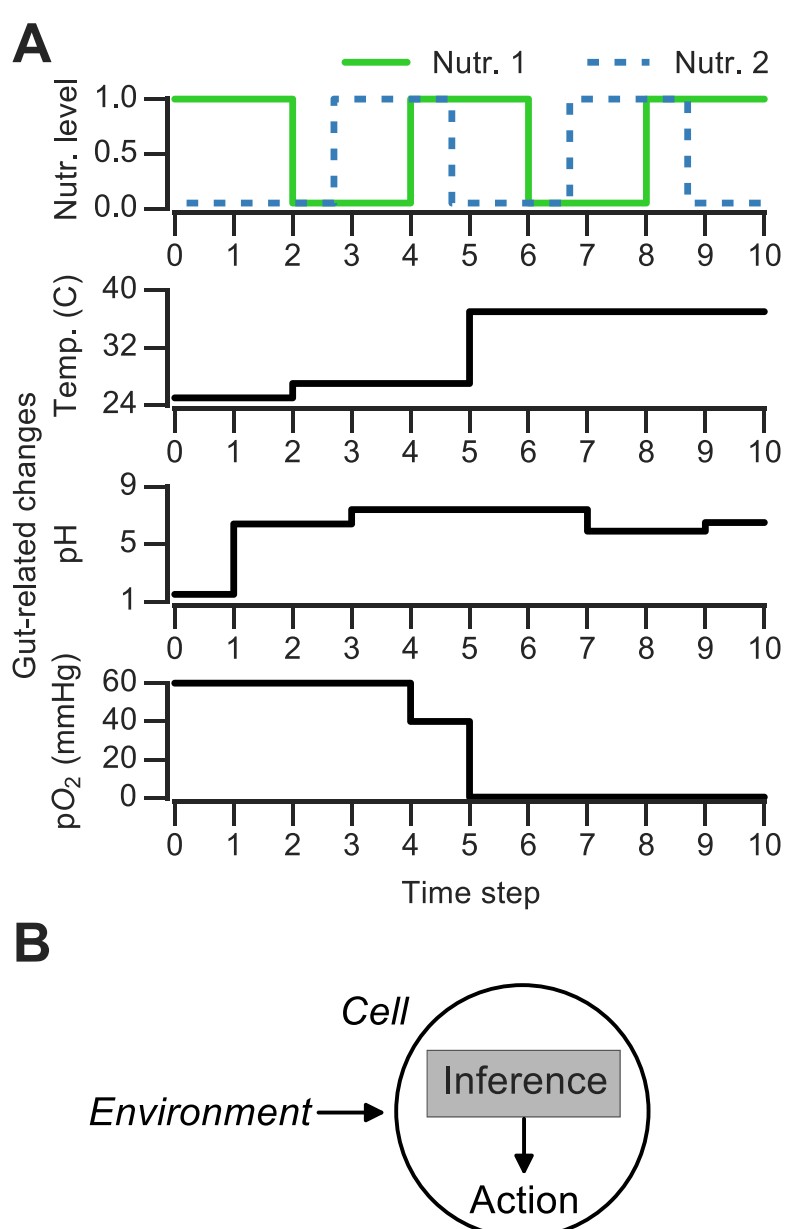

**Figure 1** **Discrete changing environments and cellular adaptation to change as inference.** (A) Examples of discrete changing environments (time in arbitrary units). Top plot indicates sharp fluctuations in two nutrients. Remaining plots show changes thought to be experienced by microbes when interacting with host gut (unrelated to top nutrient panel), which include changes in temperature (25 °C in external environment, 27 °C on human skin, 37 °C in gut), pH and oxygen levels (see main text). (B) Schematic of interaction between cell and changing environment in our framework. Cells sense dynamic environment over time, make inferences about the future state of the environment and use these predictions to take action (e.g., upregulate genes required to metabolize a nutrient).

temperature (Fig. 1A). Once in the gastrointestinal tract, microbes can face distinct pH regimes, ranging from acidic environment of the stomach (pH 1.5–5) to the intestinal duodendum (pH 5–7), jejunum (pH 7–9) and ileum (pH 7–8) to the colon (pH 5–7) (*Walter & Ley, 2011*)—these fluctuations are shown schematically in Fig. 1A. The gut

lumen also contains an oxygen gradient (*Albenberg et al., 2014*), and experimentally induced oxygenation or oxygenation as part of medical procedures (such as ileostomies) result in sharp shifts in oxygen levels that reversibly alter microbiota composition (*Hartman et al., 2009*; *Albenberg et al., 2014*). Thus, both discrete and continuous features contribute to the gut environment, and discrete environments are a useful approximation for studying the response to environmental change.

In early theoretical work on changing environments (*Levins, 1968*), Richard Levins argued that the statistical relations between signals in the fluctuating environment are central to adaptation. While distinct cell populations or strains may have different costs associated with each environmental state—e.g., distinct yeast strains exhibit different "preparation times" when switched from a glucose to a galactose environment (*Wang et al., 2015*)—the statistical properties of the environment remain informative for adaptation regardless of these costs.

Here, we develop a computational framework for characterizing the statistical structure of changing discrete environments and the adaptive strategies that support optimal growth in these environments. We focus on environments that are characterized by a blend of randomness and order, of the sort one would expect in the gut or other rich microbial ecosystems. We propose that adaptation to changing environments can be framed as inference under uncertainty (*Jaynes, 2003*). We begin with highly simplified formulation of growth in changing environments, but progressively build up to an inference-based adaptation strategy that is: (1) suited for growth in complex multi-nutrient environments, and (2) can be implemented in principle using a simple cellular circuit. Although we illustrate our results in terms of glucose-galactose adaptation in yeast, our framework applies broadly to microbial adaptation in other types of fluctuating environments.

## MATERIALS & METHODS

### Growth rate measurements

Growth rates for 61 yeast strains were measured as described in *Wang et al. (2015)*. Briefly, OD600 growth measurements were log-transformed, fit by splines and the region with maximal derivative in the spline fit ("exponential phase") was used to calculate the growth rate (defined as doublings per hour). Each strain was measured in duplicate and the average growth rate was used in Fig. S1.

### Optimal policies in Markov environments with two nutrients

In Fig. 2, the ratio of the expected growth rate obtained by following a posterior predictive policy (where the most probable nutrient under the posterior is chosen in the next step) to expected growth rate using a glucose-only policy is shown. Below, we describe in detail how this ratio was calculated.

To compare the growth rate differences between a glucose-only policy and the posterior predictive policy in two-nutrient Markov environments, we assumed an idealized case where the transition probabilities $\theta_{\text{Glu}\to\text{Glu}}$ and $\theta_{\text{Gal}\to\text{Glu}}$ are known. The "optimal" policy relative to an environment is one that maximizes the expected growth rate.

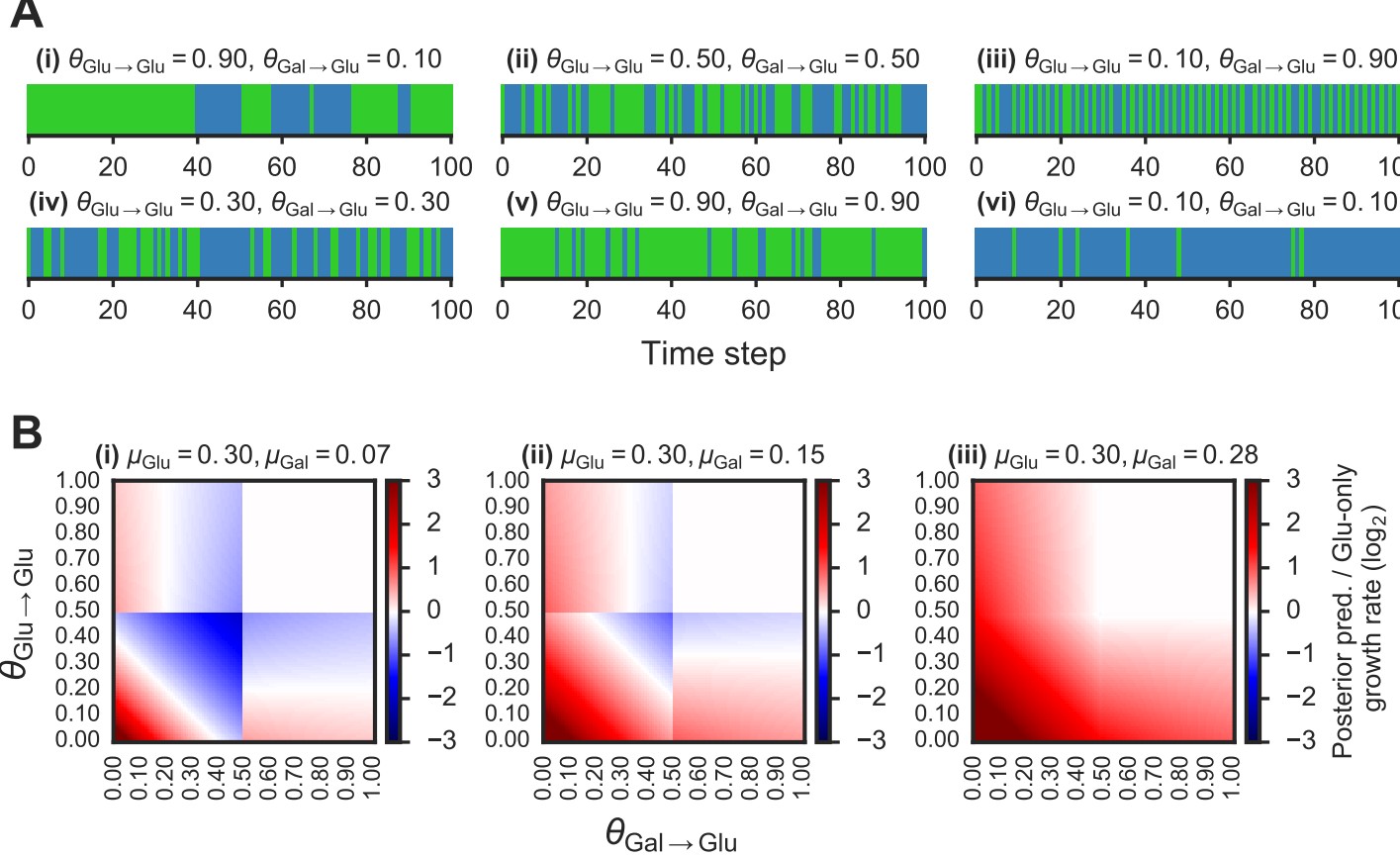

**Figure 2** **Fitness benefit of exploiting the probabilistic structure of Markov nutrient environments.** A(i)–(vi) Discrete Markov nutrient environments, characterized by two parameters: the transition probability from a glucose state back to glucose state ($\theta_{Glu \to Glu}$) and from galactose state to glucose state ($\theta_{Gal \to Glu}$). Environment assumed to switch from fixed levels of glucose to galactose, visualized as rectangles (collapsing the $y$-axis from nutrient environment such as one shown in Fig. 1A where glucose environmental state is shown in green, galactose state in blue). B(i)–(iii) Comparison of expected growth rates using the posterior predictive compared with the glucose-only strategy. Heat maps show fold-change in expected growth using posterior predictive strategy relative to glucose-only expected growth rate, as a function of the transition probabilities ($\theta_{Glu \to Glu}$ and $\theta_{Gal \to Glu}$) that parametrize the environment (see 'Materials & Methods' for detailed calculation). From (i) to (ii), increasing galactose growth rate ($\mu_{Gal}$) with fixed glucose growth rate ($\mu_{Glu}$).

The expected growth rate is dependent on the environment's transition probabilities, the growth rates afforded by each of the nutrients, as well as the cost of being "mismatched" to the environment (i.e., being tuned to a nutrient that isn't present in the environment.)

In our Markov nutrient environment, there are two environment states (glucose or galactose) and two possible actions for the cell population, each associated with a different growth rate. Notation for these states and parameters is as follows:

- Environment states: the state of the environment at time $t$ is represented by the random variable $C_t$, which takes on one of two values, $c_1 = \text{Glu}, c_2 = \text{Gal}$.
- Transition probabilities: $\theta_{Glu \to Glu}, \theta_{Gal \to Glu}$.
- Actions: tuning to glucose ($\alpha_1 = \text{Glu}$) or tuning to galactose ($\alpha_2 = \text{Gal}$).
- Growth rates associated with each action and environment state:

- When tuned to glucose in glucose environment: $V_{11}$ (identical to growth rate on glucose, $\mu_{Glu}$, from main figures)
- When tuned to glucose in galactose environment: $V_{12}$
- When tuned to galactose in galactose environment: $V_{22}$ (identical to growth rate on galactose, $\mu_{Gal}$, from main figures)
- When tuned to galactose in glucose environment: $V_{21}$.

A policy $\pi$ is a mapping from an environment's state to an action. We can now write down the *conditional growth rate* associated with a particular policy, $R(\pi|C_{t-1})$, which is the growth rate given the previous environment state $C_{t-1}$. Let $\pi_1$ and $\pi_2$ correspond to policies that constitutively tune to glucose or galactose, respectively. The conditional growth rates for these policies are:

$$R(\pi_1|C_{t-1}) = V_{11}P(C_t = c_1|C_{t-1}) + V_{12}P(C_t = c_2|C_{t-1})$$
$$R(\pi_2|C_{t-1}) = V_{22}P(C_t = c_2|C_{t-1}) + V_{21}P(C_t = c_2|C_{t-1}).$$

For simplicity, we assume that the growth rate is zero when the internal state of the cells is mismatched to the environment, i.e., $V_{12} = V_{21} = 0$. The conditional growth rates then simplify to:

$$R(\pi_1|C_{t-1}) = V_{11}P(C_t = c_1|C_{t-1})$$
$$R(\pi_2|C_{t-1}) = V_{22}P(C_t = c_2|C_{t-1}).$$

To get the expected growth rates, we sum over the possible states of the environment at time $t-1$, yielding:

$$R(\pi_1) = V_{11}\big[\theta_{Glu \to Glu} + \theta_{Gal \to Glu}\big]$$
$$R(\pi_2) = V_{22}\big[(1 - \theta_{Glu \to Glu}) + (1 - \theta_{Gal \to Glu})\big]$$

The policy $\pi_1$ is optimal when $\frac{R(\pi_1)}{R(\pi_2)} > 1$.

Unlike the glucose-only or galactose-only policy, the posterior predictive policy chooses the next action based on the transition probabilities $\theta_{Glu \to Glu}$ and $\theta_{Gal \to Glu}$. This policy, denoted $\pi_3$, chooses the most probable nutrient as a function of the environment's previous state $C_{t-1}$:

$$\pi_3(C_{t-1}) = \begin{cases} \begin{cases} \alpha_1 & \text{if } \theta_{Glu \to Glu} > 0.5 \\ \alpha_2 & \text{otherwise} \end{cases} & \text{if } C_{t-1} = c_1, \\ \begin{cases} \alpha_1 & \text{if } \theta_{Glu \to Glu} > 0.5 \\ \alpha_2 & \text{otherwise} \end{cases} & \text{if } C_{t-1} = c_2. \end{cases}$$

We can now write the expected growth rate for the posterior predictive policy by analyzing each of the cases involving the transition probabilities $\theta_{Glu \to Glu}$ and $\theta_{Gal \to Glu}$:

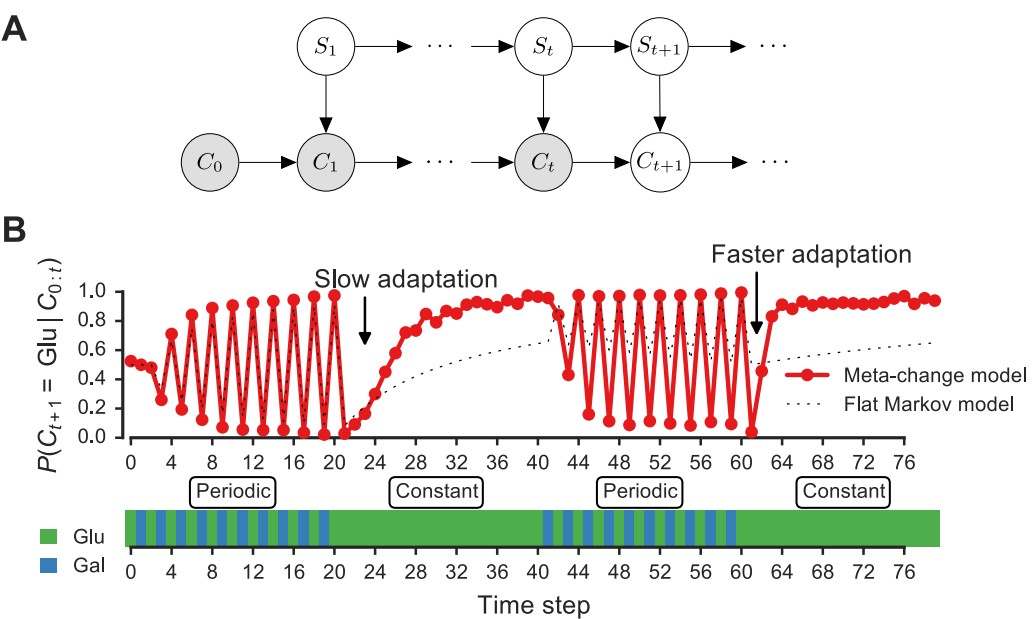

**Figure 3 Signature of adaptive behavior resulting from using the posterior predictive strategy.** (A) Dynamic Bayesian model for meta-changing nutrient environments in graphical model notation (*Jordan, 2004*): grey nodes represent observed variables, white nodes represent hidden variables. Nutrient value at time $t$, $C_t$, depends on the nutrient value at time $t-1$, $C_{t-1}$, and on the switch state $S_t$. See Fig. S2 for detailed graphical model. (B) Posterior predictive probability, obtained by particle filtering, of glucose using the full model (red), and using a Markov model with "flat" structure and no hidden states (dotted black line). In the first transition from the periodic to constant glucose environment, the posterior predictive probability in the full model updates more slowly compared to the second transition from periodic to constant environment.

$$R(\alpha_3) = \begin{cases} V_{11}\theta_{\text{Glu}\to\text{Glu}} + V_{11}\theta_{\text{Gal}\to\text{Glu}}, \\ \quad \text{if } \theta_{\text{Glu}\to\text{Glu}}, \theta_{\text{Gal}\to\text{Glu}} \geq 0.5 \\ V_{11}\theta_{\text{Glu}\to\text{Glu}} + V_{22}(1-\theta_{\text{Gal}\to\text{Glu}}), \\ \quad \text{if } \theta_{\text{Glu}\to\text{Glu}} \geq 0.5 \text{ and } \theta_{\text{Gal}\to\text{Glu}} < 0.5 \\ V_{22}(1-\theta_{\text{Glu}\to\text{Glu}}) + V_{11}\theta_{\text{Gal}\to\text{Glu}}, \\ \quad \text{if } \theta_{\text{Glu}\to\text{Glu}} < 0.5 \text{ and } \theta_{\text{Gal}\to\text{Glu}} \geq 0.5 \\ V_{22}(1-\theta_{\text{Glu}\to\text{Glu}}) + V_{22}(1-\theta_{\text{Gal}\to\text{Glu}}), \\ \quad \text{if } \theta_{\text{Glu}\to\text{Glu}}, \theta_{\text{Glu}\to\text{Glu}} < 0.5. \end{cases}$$

The ratio $\frac{R(\pi_3)}{R(\pi_1)}$ is shown in Fig. 2 as a function of $\theta_{\text{Glu}\to\text{Glu}}$ and $\theta_{\text{Gal}\to\text{Glu}}$ for different values of each nutrient's growth rate ($V_{11}$ and $V_{22}$).

## Bayesian model for meta-changing nutrient environments

Here we describe in detail the dynamic Bayesian model used to adapt to meta-changing environments (such as the one shown in Fig. 3). This model is in the family of dynamic probabilistic models, also called "switching state space models," that have been widely used in artificial intelligence, robotics and machine learning (*Ghahramani & Hinton, 1998*; *Murphy, 2002*). In our model, there are $K$-many hidden switch states that are used to produce one of $J$-many "outputs." The outputs are the different nutrients and

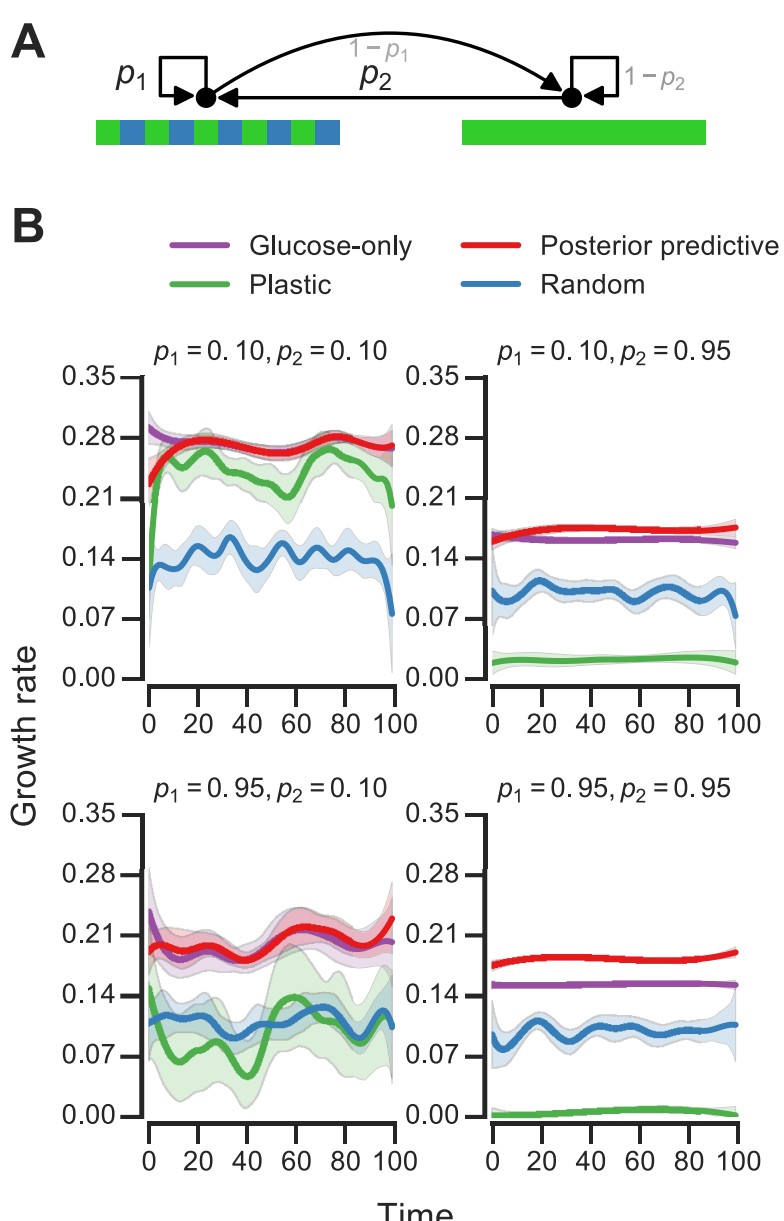

**Figure 4** **Fitness of growth policies in meta-changing environments.** (A) Meta-changing environment with two hidden states: a periodic environment and a constant environment. The hidden state transitions are parameterized by the probability of transitioning from the periodic environment to itself, $p_1$, and from the constant to the periodic environment, $p_2$. (B) Growth rates obtained by growth using different policies in meta-changing environments (mean growth rates from 20 simulations plotted with bootstrap confidence intervals as shaded regions). Four environments shown, each parameterized by different settings of $p_1, p_2$ (where $\mu_{Glu}$ is twice $\mu_{Gal}$). Posterior predictive policy generally outperforms other policies. Mean growth rates from 20 simulations plotted with bootstrap confidence intervals (shaded regions).

the switch states correspond to stretches of the environment such as the "periodic" or "constant" regions shown in Fig. 4A. Each switch state represents a transition matrix that is used to generate nutrient fluctuations. The nutrient produced at time $t$, $C_t$, depends on the value of the switch state $S_t$ and the previous nutrient $C_{t-1}$. The next switch state

$S_{t+1}$ is generated conditioned on $S_t$ based on a separate set of switch state transition probability, then $C_{t+1}$ is generated conditioned on $S_{t+1}$ so on.

The transition probabilities associated with each switch state, as well as the probabilities of transitioning between switch states, all have to be learned from the environment. We therefore place a prior on these transition probabilities.

The full graphical model including hyperparameters is shown in Fig. S2 using plate notation (*Jordan, 2004*). Formally, a switch state takes on one of $1, \ldots, K$ values. The initial switch state $S_1$ is drawn from a probability distribution on the initial switch state values, $\pi_{s_1}$. Since $\pi_{s_1}$ is unknown, we place a prior on it using the Dirichlet distribution, i.e.,:

$$\pi_{s_1} \sim \text{Dirichlet}(\alpha_{s_1})$$
$$S_1 | \pi_{s_1} \sim \text{Multinomial}(\pi_{s_1}, 1)$$

which means $P(S_1 = i | \pi_{s_1}) = \pi_{s_1}^{(i)}$, where $\pi_{s_1}^{(i)}$ denotes the $i$th entry in $\pi_{s_1}$. For clarity, we will sometimes omit the explicit value assignment for a random variable, and write $P(S_t = i | \pi_{s_1})$ as simply $P(S_1 | \pi_{s_1})$. The switch states at later time points are generated as follows: each switch state $i \in \{1, \ldots, K\}$ has an associated probability vector $\pi_i$, whose $j$th entry $\pi_i^{(j)}$ determines the probability of transitioning from switch state $i$ to $j$. The probability of the switch state at time step $t > 1$ therefore depends on the previous switch state's value $S_{t-1}$ and the switch state transition probabilities:

$$\pi_i \sim \text{Dirichlet}(\alpha_s)$$
$$S_t | S_{t-1} = i \sim \text{Multinomial}(\pi_i, 1).$$

Similarly, each nutrient state can take one value $i \in \{1, \ldots, J\}$, and the initial nutrient state $C_0$ is drawn from a probability distribution, $\pi_{c_0}$, which is in turn drawn from a Dirichlet prior distribution:

$$\pi_{c_0} \sim \text{Dirichlet}(\alpha_{c_0})$$
$$C_0 | \pi_{c_0} \sim \text{Multinomial}(\pi_{c_0}, 1).$$

The probability of a nutrient output at time $t > 0$ depends on the previous nutrient $C_{t-1}$ and on the switch state at time $t$. The switch state value indexes which transition probability distribution to use for the outputs, and as before, the transition probabilities are drawn from a Dirichlet prior:

$$\pi'_{i,j} \sim \text{Dirichlet}(\alpha_c)$$
$$C_t | C_{t-1} = i, S_t = j \sim \text{Multinomial}(\pi'_{i,j}, 1).$$

The posterior predictive distribution, $P(C_{t+k} | C_{0:t})$, is the main quantity of interest (we assume $k = 1$ as in all of our simulations). This distribution can be calculated recursively in dynamic probabilistic models, a property that we will use in a later section to derive a real-time estimation procedure for this distribution using particle filtering. (For an accessible introduction to recursive estimation of Bayesian posteriors and to

particle filtering, see Ch. 1 in *Murphy (2002)*, Ch. 1-3 in *Särkkä (2013)* or *Cappé, Godsill & Moulines (2007)*). We derive the posterior predictive distribution in steps. First, consider the posterior distribution over the switch state at time $t$ given the environment history up until and including time $t$, $P(S_t|C_{0:t})$, called the *filtering posterior* distribution, which is obtained by marginalizing out the switch state $S_{t-1}$:

$$P(S_t|C_{0:t}) \propto P(C_t|S_t) \sum_{S_{t-1}} P(S_t|S_{t-1}) P(S_{t-1}|C_{0:t-1}).$$

Note that the third term is the filtering posterior over the switch state at time $t-1$, which can be rewritten as we did above in terms of the filtering posterior at $t-2$, and so on—this shows that the posterior can be computed recursively. The base case of the recursion is given by our prior distributions over the initial nutrient state $C_0$ and the initial switch state $S_1$ (as shown in Fig. S2).

The posterior predictive distribution $P(C_{t+1}|C_{0:t})$ can then be written as a product that uses the filtering posterior, marginalizing out the hidden switch states:

$$P(C_{t+1}|C_{0:t}) \propto \sum_{S_t} \sum_{S_{t+1}} P(C_{t+1}|C_t, S_{t+1}) P(S_{t+1}|S_t) P(S_t|C_{0:t-1}).$$

The distributions that the filtering posterior decomposes to depend on parameters that are unobserved, such as the transition probabilities. Since Dirichlet-Multinomial distributions are conjugate (*Gelman et al., 1995*), we can analytically integrate out these parameters. For example, the transition probabilities $\pi$ for the switch states can be integrated out:

$$P(S_{t+1}|S_t = i) = \int P(S_{t+1}|S_t = i, \pi_i) P(\pi_i|\alpha_s) \mathrm{d}\pi_i.$$

Similarly, the probability of observing a nutrient given the previous nutrient and the switch state, $P(C_{t+1}|C_t, S_{t+1})$, can be computed while integrating out the nutrient transition probabilities $\pi'_{i,j}$:

$$P(C_{t+1}|C_t = j, S_{t+1} = i) = \int P(C_{t+1}|C_t = j, S_{t+1} = i, \pi'_{i,j}) P(\pi'_{i,j}|\alpha_c) \mathrm{d}\pi'_{i,j}.$$

We discuss in detail how to estimate these distributions from observations in the section on particle filtering.

*Values of hyperparameters:* In all analyses, we set hyperparameters $\alpha_{s_1}, \alpha_{c_0}, \alpha_c$ to be vectors consisting of all ones. The hyperparameter on switch state transitions $\alpha_s$ was set such that self-transitions get the hyperparameter value 2, i.e., for the $i$th switch state, $\alpha_s^{(i)} = 2$, and all other entries in $\alpha_s$ are set to 1. This encodes a weakly "sticky" prior that slightly favors self-transitions for hidden switch states.

## Real-time inference using particle filtering

We estimate the posterior predictive distribution above in real-time using particle filtering (*Cappé, Godsill & Moulines, 2007*) as outlined in Algorithm S1. Particle filtering for our

model is implemented in the particlefever library (available on Github.) We used 200 particles for all simulations with particle states initialized from the prior.

In particle filtering, a distribution of hidden state values (in our model, the hidden switch states) is represented using a set of particles. Each particle corresponds to a configuration of the hidden states (configurations are typically assigned from the prior distribution at time $t = 0$). The particles are assigned weights that are initialized to be uniform. Starting with time $t$, the particle filtering algorithm works by first *predicting* hidden state values particles $t + 1$. When a data point at time $t + 1$ is observed, the weight of each particle is *updated* to be proportional to the likelihood of the new data point given the particle's hidden state configuration. This process is repeated as data points are observed. To prevent particle degeneracy (a case where particles get weights that are too low), a resampling step is used where particles are sampled in proportion to their weights and the weights are reset to be uniform.

We now turn to the representation of our state space that is encoded in each particle. As discussed above, because of the conjugacy of Dirichlet-Multinomial distributions (*Gelman et al., 1995*), we can analytically integrate out the transition probabilities in our model. This means that these transition probabilities don't have to be represented in our particles. Instead, the sufficient statistics' for our model are simply: (1) a matrix of counts $\mathbf{S}$ where $\mathbf{S}^{(i,j)}$ is the number of times hidden switch state $i$ transitioned to hidden state $j$ in the particle's trajectory, and (2) a three-dimensional array of counts $\mathbf{C}$ where $\mathbf{C}^{(i,j,k)}$ is the number of times nutrient $i$ transitioned to nutrient $j$ under switch state $k$. The predict/update cycles for a particle $p = \{s_t, \mathbf{S}, \mathbf{C}\}$, where $s$ is the particle's hidden state at time $t$, are:

*Prediction step:* For each particle $p$ we draw a new switch state for $t + 1$, $s_{t+1} \sim P(S_{t+1}|S_t = s_t)$. By conjugacy, the switch state transition probabilities $\pi_{s_t}$ can be integrated out, yielding the posterior predictive distribution for a Dirichlet-Multinomial:

$$P(S_{t+1} = s_{t+1}|S_t = s_t) \propto \frac{\alpha_s + \mathbf{S}^{(s_t, s_{t+1})}}{\sum_k (\alpha_s + \mathbf{S}^{(s_t, k)})}$$

which can be sampled from.

*Updating step:* When a nutrient $c_{t+1}$ is observed, we update the weight $w$ of our particle $p = \{s_{t+1}, \mathbf{S}, \mathbf{C}\}$ in proportion to $P(C_{t+1}|C_t, S_{t+1})$:

$$w \propto P(C_{t+1} = c_{t+1}|C_t = c_t, S_{t+1} = s_{t+1}).$$

Integrating out the nutrient transition probabilities $\pi'_{c_t, c_{t+1}}$, this also gives the Dirichlet-Multinomial posterior predictive distribution:

$$w \propto \frac{\alpha_c + \mathbf{C}^{(c_t, c_{t+1}, s_{t+1})}}{\sum_k (\alpha_c + \mathbf{C}^{(c_t, k, s_{t+1})})}.$$

See Algorithm S1 for remaining details.

### Fitness simulations

We simulated growth with different policies using a simple model of exponential growth. Cells were assumed to grow exponentially with a growth rate determined by the environment's nutrient state. The initial population size and the time duration of each environment simulated are as described in figure legends. To determine growth rates empirically, we fitted splines to the log of the population sizes across the time and computed the first derivative. All code for fitness simulations is available in the paper's Github repository.

### Molecular implementation of nutrient transition counter

The nutrient transition counter model was drawn in CellDesigner (version 4.4) (*Funahashi et al., 2008*), serialized as an SBML file, and simulated in Python using libRoadRunner. The SBML file for the model is available at 10.6084/m9.figshare.3493994. A detailed report of the chemical reactions and rate parameters was generated by SBML2 LATEX (*Drager et al., 2009*) and is available at 10.6084/m9.figshare.3492185.

## RESULTS

### Growth advantage of using the environment's probabilistic structure

We first asked whether an adaptive strategy, which exploits the probabilistic structure of the environment, would pay off in terms of growth compared with a non-adaptive strategy. We considered a class of *Markov nutrient environments* that fluctuate between two nutrients, glucose and galactose, where nutrient changes follow a Markov model. The probabilistic structure of these environments is determined by two parameters: the probability of transitioning from glucose to glucose ($\theta_{\mathrm{Glu}\to\mathrm{Glu}}$) and the probability of transitioning from galactose to glucose ($\theta_{\mathrm{Gal}\to\mathrm{Glu}}$), as shown in Fig. 1A. (Equivalently, the model can be parameterized by $\theta_{\mathrm{Glu}\to\mathrm{Gal}}$ and $\theta_{\mathrm{Gal}\to\mathrm{Glu}}$, since $\theta_{\mathrm{Glu}\to\mathrm{Gal}} = 1 - \theta_{\mathrm{Glu}\to\mathrm{Glu}}$.) Intuitively, the higher $\theta_{\mathrm{Glu}\to\mathrm{Glu}}$ and the lower $\theta_{\mathrm{Gal}\to\mathrm{Glu}}$, the more likely we are to encounter glucose in the environment. Different settings of the transition probabilities can produce qualitatively different environments (Fig. 2A). We also assume that one nutrient (in this case, glucose) confers a higher growth rate than alternative nutrients, which is typically the case.

In our framework, the behavior of a cell population is determined by a *policy*: a mapping from the environment's past state to a future action. We compared the performance of two policies in Markov environments: a *posterior predictive* policy, in which cells tune to the most probable nutrient (based on $\theta_{\mathrm{Glu}\to\mathrm{Glu}}$ and $\theta_{\mathrm{Gal}\to\mathrm{Glu}}$), and a non-adaptive strategy in which cells are constitutively tuned to the preferred nutrient, glucose. The quantity of interest in the posterior predictive policy is the posterior predictive distribution, which is the probability of a nutrient at the next time step given the previously observed nutrients: $P(C_{t+1}|C_{0:t})$, where $C_{t+1}$ denotes the nutrient at time $t+1$ and $C_{0:t}$ denotes the environment's nutrient history up until $t$. The posterior predictive policy chooses the nutrient that maximizes this probability (see 'Materials & Methods' for details).

To compare the fitness difference between these policies, we simulated population growth with each policy using a highly simplified growth model, similar to one used in *Jablonka et al. (1995)*. In this model, we assume that cells respond to changes in the environment with a delay, or "lag." Cells tune to a nutrient at time $t$ and incur a change in growth rate as a consequence of this decision at a later time $t + k$, where $k$ is the lag parameter (we assume here that $k = 1$). For growth kinetics, we assume that: (1) cells grow exponentially when their nutrient state matches the environment's state, (2) there is no switching cost for cells between nutrient states, and (3) when cells are "mismatched" to their environment—i.e., tuned to a nutrient that is not present—their growth rate is zero (this assumption is supported by the observation that cells lacking Gal4, a transcription factor required to activate the GAL pathway, cannot grow in galactose alone (*Escalante-Chong et al., 2015*)). Altogether, our growth assumptions represent extreme conditions, but they serve as a useful starting point for seeing when an adaptive strategy is beneficial to population growth.

Given these growth assumptions, we plotted for each Markov environment the ratio of expected growth rate using the posterior predictive policy to the expected growth rate under the glucose-only policy in Fig. 2B (the detailed calculation of these ratios is described in 'Materials & Methods'). To constrain our choice of growth rates, we analyzed growth measurements of 61 yeast strains cultured with different sugars as primary carbon sources (see 'Materials & Methods' for details). As expected, median growth rate in glucose was higher than in other sugars (Fig. S1A). Across strains, growth rate in glucose was on average ∼1.5 fold higher than in galactose and in some strains over 3-fold higher (Fig. S1B).

We find that in environments where glucose yields a significantly larger (∼2–4 fold) growth rate than galactose, the posterior predictive policy outperforms the glucose-only policy only in specific regimes of the space of possible Markov environments (red regions in Fig. 2B). For a wide range of environments, the non-adaptive policy is equal to or better than the adaptive policy. As expected, when the difference in growth rate between glucose and galactose gets smaller (Fig. 2B, left to right), the payoff from using the posterior predictive policy is greater. However, as our analysis of growth rates in yeast strains showed, glucose typically confers a substantially higher growth rate than galactose.

This idealized calculation shows the importance of the probabilistic structure of the environment in assessing where an adaptive strategy would pay off. This suggests that in order to highlight the advantage of adaptive strategies, specific types of environmental fluctuations will have to be used.

## Adapting to meta-changing environments with real-time inference

Our analysis of Markov nutrient environments above doesn't capture several key aspects of adaptive growth in changing environments. First, our environment's changes had a simple "flat" structure describable by only two parameters, whereas natural environment may be generated by far more complex underlying mechanisms. Second, our comparison of the adaptive and glucose-only strategies assumed that the transition probabilities governing the environment, $\theta_{Glu \rightarrow Glu}$ and $\theta_{Gal \rightarrow Glu}$, are known and can be used by

cells. In reality, if this information can be used by cells, it has to be learned from the environment and cannot be assumed as given. Third, such information has to be learned in real-time as cells must respond to the environment while it is changing. We now address each of these aspects in turn.

Changes in complex environments may be governed by dynamic processes that are unobservable to cells. Some environments may oscillate between noisy regimes, where the nutrient switches are less predictable, and stable regimes where the nutrient switches are either rare or more predictable. Such environments can be thought of as "meta-changing" in the sense that there's a change in the way they fluctuate: the probability of being a specific state of the environment (e.g., where a nutrient is available) changes through time, depending on an unobserved condition, such as whether we're in a noisy or stable regime. The transition from noisy to stable regimes might itself be governed by another time-varying mechanism. As an intuitive example of meta-changing environments, consider the eating routine of animals like us. During feeding, bursts of nutrients that are otherwise scarce may become available in the gut. The fluctuations in nutrient levels within a feeding period will depend on what and how much is being consumed. The separation between meals is also subject to randomness, but can still be predictable, depending on how consistent we are in our eating schedule. This high-level structure can be exploited by adaptive systems to anticipate future changes and to separate noisy fluctuations in nutrients from signals of feeding periods.

To understand the adaptive strategies that may be used for effective growth in such environments, we developed a dynamic Bayesian model of meta-changing environments. In our model, we assume a fixed number of hidden "switch states" that correspond to regimes in the environment, and these states are used to generate the fluctuations in nutrients (Fig. 3A—see 'Materials & Methods' for full model description.) The switch states and their dynamics are not observable, and therefore have to be learned from the environment. To do this, our model has a prior distribution over the dynamics of the unobservable switch states and the nutrient transitions in the environment ('Materials & Methods'). Through experience with the environment, these priors are updated using Bayesian inference to learn the dynamics that drive nutrient fluctuations. Formally, the goal is to compute the posterior predictive distribution over nutrients, $P(C_{t+1}|C_{0:t})$, which depends on the nutrient history $C_{0:t}$ and on the hidden switch state $S_{t+1}$:

$$P(C_{t+1}|C_{0:t}) = \sum_{S_{t+1}} P(C_{t+1}|C_t, S_{t+1}=i)P(S_{t+1}|C_{0:t}). \tag{1}$$

For an inference-based strategy to be biologically plausible, it has to be carried out in real-time since cells respond to the environment while it is changing. To compute the posterior predictive distribution (Eq. (1)) in real-time, we use a particle filtering algorithm (*Cappé, Godsill & Moulines, 2007*) (described in detail in Algorithm S1 and 'Materials & Methods'). Rather than naively "memorizing" the environment's history, in particle filtering a representation of the posterior distribution is continuously updated as the environment changes ('Materials & Methods'). The uncertainty of the distribution

is represented by a set of "particles" (which can be thought of as samples from a distribution). Each particle corresponds to a configuration of the hidden states of our system. For our model, each particle $p$ is a set $\{s, \mathbf{S}, \mathbf{C}\}$, where $s$ corresponds to the value of the hidden switch state, $\mathbf{S}$ is a transition matrix that tracks the frequencies of transitions between switch states and $\mathbf{C}$ is a transition matrix that tracks the frequency of transitions between nutrients ('Materials & Methods').

The particle filtering algorithm can be understood by analogy to evolution through mutation and selection. Initially, all particles are weighted equally. Before the environment changes, particles are probabilistically assigned to new configuration based on our model of the environment ("mutation" step). When a new state of the environment is observed, the particles are re-weighted by their fit to this observation and probabilistically resampled using the updated weights ("selection" step). This process repeats as the environment continues to change. Particles that represent more probable states of the environment will get "selected" for through time, while the "mutation" and resampling steps ensure that diversity is brought into the particle population. As each particle $p$ goes through this process, the hidden state and nutrient state configurations in its trajectory are counted by updating the transition matrices $\mathbf{S}$ and $\mathbf{C}$. Since the particles are propagated through this process in parallel, particle filtering takes what naively would be a serial computation (requiring the complete nutrient history) and converts it to a parallel one that can be performed in real-time. This property may make particle filtering amenable to implementation by molecular circuits, as discussed later.

## Signatures of Bayesian adaptation in meta-changing environments

Our model makes a number of predictions about the dynamics of adaptation by systems that represent hidden environmental states. In Fig. 3B, a meta-changing environment is shown that switches between two regimes: one with periodic switches between glucose and galactose, and another where glucose is constant. The posterior predictive probability of glucose in the next time step, $P(C_{t+1}|C_t)$, as it gets updated by real-time inference, is plotted along the environment (Fig. 3, top).

The change in the posterior predictive distribution has a number of characteristic features. Starting with a uniform probability over the nutrients, the posterior predictive distribution slowly changes to "learn" the first periodic regime of the environment (Fig. 3B, red line). When the environment shifts to the constant regime (at $t = 20$, Fig. 3B), the posterior distribution also updates slowly to reflect this. However, when the environment shifts again to the periodic regime (at $t = 40$), the posterior predictive distribution updates faster to reflect the periodicity. Similarly, when the environment shifts to the constant region for the second time (at $t = 60$), the posterior predictive distribution changes even more quickly, since the hidden switch state where the environment is periodic has been seen before. More generally, our model predicts that the more familiar stretches of the environment will be adapted to more quickly. By contrast, a Markov model with a "flat" structure that only tracks transition probabilities between glucose and galactose does not show this behavior (Fig. 3B, dotted line). This is one of several subtle

predictions about the dynamics of adaptation one would expect from an inference-based strategy that uses a representation of hidden environmental states.

## Posterior predictive adaptation confers growth advantage in meta-changing environments

We next asked how beneficial the adaptive patterns that result from representing hidden environmental states (of the sort shown in Fig. 3) would be for growth. We compared the inference-based growth policy to other policies in meta-changing environments. We considered meta-changing environments that fluctuate between two hidden states: one where there is periodic switching between glucose and galactose, and another where glucose is constant in the environment (Fig. 4A). Each hidden state corresponds to a Markov environment (parameterized by a pair of transition probabilities, as described earlier). The switching dynamics between the hidden states are controlled by two transition probabilities, which we labelled $p_1$ and $p_2$ in Fig. 4A. We compared the growth rates obtained by using the posterior predictive policy to that obtained by plastic, random and glucose-only policies. In the "plastic" policy (described in *Jablonka et al., 1995*), cells tune to the environmental condition that they experienced last, whereas in the random policy the decision is made uniformly at random. We found that across different settings of $p_1$ and $p_2$, the posterior predictive policy generally results in faster (in some regimes, nearly two-fold higher) growth rates, which are often less variable, compared with other policies (Fig. 4B).

## A bet-hedging growth policy based on real-time inference

While all the policies we have considered so far act at the population-level, bet-hedging has been proposed as an adaptive strategy in fluctuating environments (*Veening, Smits & Kuipers, 2008*). With bet-hedging, different fractions of the population are tuned to different environment conditions, and these cellular states can be inherited through several generations through epigenetic mechanisms (*Veening, Smits & Kuipers, 2008*). Previously, a bet-hedging policy was shown in simulation to give increased fitness when the bet-hedging proportions matched the frequencies of the environment's fluctuations (*Jablonka et al., 1995*) (in that study, bet-hedging was referred to as "carry-over"). However, as we discussed above, there is no way for a population of cells to know these probabilities in advance. Our real-time inference scheme lends itself to a bet-hedging policy where the fraction of cells devoted to an environmental state is determined by the posterior predictive probability of this state, which is learned in real-time. Using a growth fitness simulation in the same meta-changing environments, we found that a bet-hedging posterior predictive policy performs similarly to its population-level counterpart and outperforms a random bet-hedging policy (Fig. S2). Real-time probabilistic inference therefore gives us a principled approach for adaptively setting bet-hedging proportions.

## Adapting to multi-nutrient meta-changing environments

In natural nutrient environments, unlike in most laboratory conditions, multiple nutrients that can be metabolized by cells may be available. We next asked how distinct growth strategies would do in such multi-nutrient environments.

We analyzed Markov environments with three nutrients—glucose, galactose and maltose—that many yeast strains can grow on as primary carbon sources (as shown in Fig. S1). Three qualitatively distinct environments generated by varying the transition probabilities that control these environments are shown in Fig. 5A. The first environment has "persistent" stretches of each of the nutrients, the second is a mixture of periodic glucose-galactose switches and persistent maltose states, and the third is one where the presence of glucose signals an upcoming stretch of galactose, while galactose signals an upcoming persistent maltose stretch (Fig. 5). Even with only three nutrients, qualitatively rich environments such as these can be constructed.

For fitness comparisons of growth policies, we considered meta-changing environments that switch between the second and third multi-nutrient environments shown in Fig. 5A. These switches, as in previous meta-changing environments, are governed by two transition probabilities, $p_1$ and $p_2$ (Fig. 5B, top). It's straightforward to apply the posterior predictive policy to multi-nutrient environments by running real-time inference using a version of our Bayesian model that assumes three rather than two nutrient states. We compared the growth performance of this posterior predictive policy to that of the plastic, random and glucose-only policies for different settings of $p_1$ and $p_2$ (Fig. 5B). In all cases, the posterior predictive policy results in growth rates that are larger than those produced by other policies.

## Molecular circuit design for inference-based adaptation

Our results suggest that cell populations that use a probabilistic inference-based growth strategy can achieve greater fitness, but it's not clear how such a strategy can be realized in molecular circuits. We next asked whether the particle filtering algorithm can inform the design of molecular circuits that implement the posterior predictive strategy.

In order to implement the posterior predictive strategy, we need a way to encode the inference algorithm's representation of the environment in molecules. Our particle filtering algorithm uses a minimal representation of (Markovian) environments that a circuit that performs inference in real-time would have to track. This representation consists of the hidden switch state $s$, and a pair of transition matrices $\mathbf{S}$ and $\mathbf{C}$ that track hidden state and nutrient state transitions, respectively (as described above and in 'Materials & Methods'). When the environment changes, these representations are updated. At the molecular level, binary state such as the hidden switch state $s$ can be represented easily using phosphorylation, protein dimerization or other known molecular switches. The more challenging part of implementing inference is keeping track of the environment's changes. As the environment changes, our inference algorithm tracks the number of times the environment transitioned from one nutrient state to another, effectively keeping a counter. Recall that the Markov environments we analyzed can be parameterized by the frequency of transitions from glucose to galactose, and from galactose to glucose. Therefore, our circuit would have to distinguish glucose-to-galactose from galactose-to-glucose switches and "count" these switches as the environment fluctuates.

To build this circuit, we need three types of components: (1) sensors that detect the presence of nutrients, (2) activators that act downstream of the sensors to turn on the

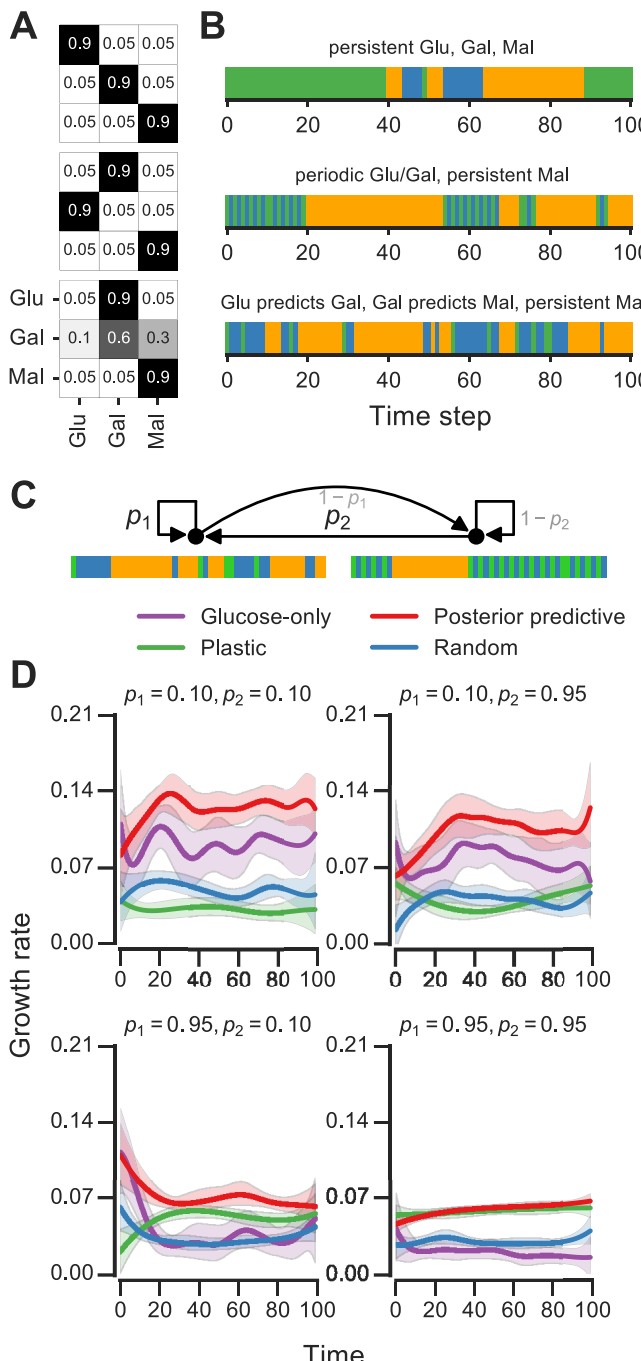

**Figure 5 Fitness of growth policies in multi-nutrient meta-changing environments.** (A, B) Three multi-nutrient Markov environments where glucose, galactose and maltose fluctuate. Transition probability matrices shown as heat maps (A) along with the environments they produce starting with glucose as initial state (B). (C) A meta-changing multi-nutrient environment that switches between the second and third Markov environments shown in (B). (D) Growth rates obtained using different policies in meta-changing environment shown in top, shown for four different settings of $p_1, p_2$. Growth rate settings used: $\mu_{Glu}$ was twice $\mu_{Gal}$ and $\mu_{Gal} = \mu_{Mal}$. Mean growth rates from 20 simulations plotted with bootstrap confidence intervals (shaded regions).

relevant metabolic pathway, and (3) "memory molecules" that record each relevant transition (glucose-to-galactose or galactose-to-glucose). Two of these component types, the sensors and the activators, are already part of the basal nutrient signaling pathway. What's left is to wire these components to the memory molecules so that the circuit can count nutrient transitions.

We constructed such a nutrient transition counter for an environment that has glucose and galactose. The eight-component circuit is shown in Fig. 6A and its reaction equations are listed in Table S1 (all parameters used in reactions are given in 'Materials & Methods'). The circuit has sensors that are activated by sugars, activators that are produced downstream of the sensors, and proteins that count glucose-to-galactose and galactose-to-glucose transitions (and act as "memory molecules"). In the left branch of the circuit, glucose catalytically activates a galactose sensor (Fig. 6A). When galactose is present, the galactose sensor reversibly forms a galactose activator ($Gal\_Sensor + Gal \rightleftharpoons Gal\_Activator$). Similarly, in the right branch of the circuit, galactose catalytically activates a glucose sensor, which in the presence of glucose reversibly forms a glucose activator ($Glu\_Sensor + Glu \rightleftharpoons Glu\_Activator$). The glucose activator triggers the production of a glucose-to-galactose transition counting molecule, while the galactose activator produces a galactose-to-glucose transition counting molecule. The transition counting molecules are assumed to have a very slow degradation rate, and this rate determines the stability of the counter's "memory."

We simulated the behavior of this circuit in an environment that switches between glucose and galactose (Fig. 6B, top). The glucose sensor is active during galactose pulses and the galactose sensor is active during glucose pulses. When the environment first switches from glucose to galactose (at $t = 50$, Fig. 6B), the galactose activator is formed and triggers a spike in the glucose-to-galactose counter (Fig. 6B, bottom). When the environment switches from galactose to glucose, the galactose-to-glucose counter spikes (at $t = 100$). The counter molecules are highly stable, so their level forms the "memory" of these two transitions. When the environment switches from glucose to galactose for the second time (at $t = 150$), the glucose-to-galactose counter spikes again to a level roughly twice that of the galactose-to-glucose counter, while the galactose-to-glucose counter is unaffected. After all nutrient switches, the circuit retains that 2 glucose-to-galactose and 1 galactose-to-glucose transitions have been observed. This information can be used downstream to implement an inference-based adaptive strategy, like the posterior predictive strategy.

A key feature of this circuit architecture is that sensors associated with one nutrient (e.g., glucose) get activated by other nutrients (such as galactose). This "crosstalk" between the two arms of the pathway enables the environmental change tracking that is needed for inference. It may be argued that it's inefficient for organisms to express a sensor for a nutrient that isn't present, but yeast cells in fact do so: for instance, cells grown in galactose express glucose sensors and transporters, while cells grown in glucose also express galactose transporters and internal galactose sensors. Our circuit design shows that a relatively simple change in wiring among mostly existing components (such as a nutrient sensor and activator), in addition to a memory molecule, is sufficient to make the

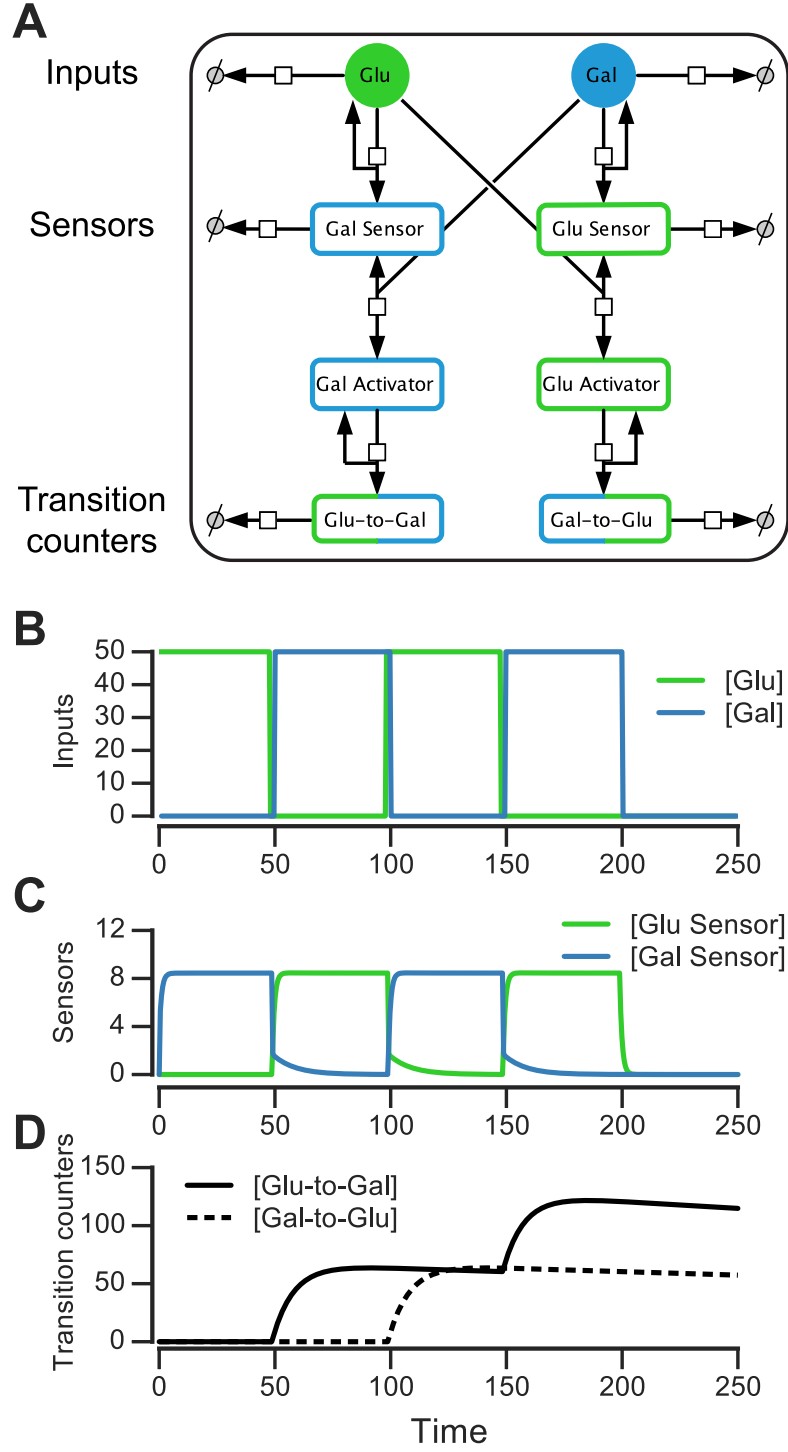

**Figure 6** **Molecular circuit implementation of a nutrient transition counter.** (A) Network of chemical reactions for implementing nutrient transition counter. ∅ denotes null species in degradation reactions. Reaction equations represented by the network are listed in Table S1. Split line arrows represent catalytic reactions, straight lines with arrowheads on both sides represent reversible reactions. (B) Input doses of glucose and galactose. (C, D) Simulation of circuit component concentrations that result from inputs shown in (B). (Concentrations are in A.U.)

transition counter. While a true digital counter is unbounded, this molecular counter's dynamic range and reliability is limited by the degradation rates and dynamic ranges of the molecular components involved (such as the sensors and counting molecules in Fig. 6A).

Whether such a circuit is likely to be used by an organism will depend on the type of fluctuations in the organism's environment and on the fitness advantage conferred by tracking metabolites (relative to the cost of tracking). These tradeoffs are currently unknown, but can be studied experimentally by engineering synthetic circuits such as the one we have proposed into cells and analyzing their fitness in various environments. It's plausible that even if such circuits exist in nature, only a subset of the nutrients cells consume may be tracked in this way.

Our circuit is a proof-of-concept design of the core machinery needed for real-time inference in our probabilistic model, but a full implementation of inference would require integration with the remaining basal glucose/galactose signaling network (as well as careful analysis of the circuit's robustness and precision).

## DISCUSSION

Fluctuations in complex environments, such as the gut, can be driven by mechanisms that cells cannot sense directly. The main contributions of this work have been to: (1) provide a framework for characterizing the computational (or information-processing) problem that cells face when living in such environments (conceived here as a form of probabilistic inference), (2) suggest particle filtering as one class of algorithms that cells may use to solve inference in real-time, and (3) propose a proof-of-concept design of a circuit that implements part of this algorithm using familiar protein biochemistry. Together, this gives an outline of a three-level analysis, following Marr's framework (*Marr, 1982*), of microbial growth in complex environments.

We found that a growth strategy based on inference, where hidden environmental features are represented, can give cells a fitness advantage. An important future direction would be to test if signatures of adaptation by inference (such as those in Fig. 3B) can be observed experimentally in glucose-galactose switching with yeast (*Stockwell, Landry & Rifkin, 2015*) or glucose-lactose switching with bacteria (*Lambert & Kussell, 2014*).

Although we assumed in our fitness simulations that the goal is to maximize population-level fitness, other goals—like minimizing the probability of population extinction (*Lewontin & Cohen, 1969*)—can be more relevant in some environments, especially for small population sizes, and these should be investigated further. Another limitation of our analysis is the assumption that environmental fluctuations follow a Markov process; an assumption violated by many natural time-varying processes. However, dynamic Bayesian models similar to the one presented here have been extended to handle non-Markov environments (*Yu, 2010*). It will be fruitful to experiment with these models and compare their assumptions to the statistical properties of natural microbial environments.

Another future challenge is to link the continuous features of the environment (which can be clearly sensed by microbes) to more abstract discrete structure like that of meta-changing environments. Elegant work by Sivak and Thomson derived optimal enzyme induction kinetics for the noisy statistics of an environment with continuously varying nutrients (*Sivak & Thomson, 2014*). This suggests that in an ideal adaptive system, principles of optimal inference are at work at multiple layers—from the abstract computational problem of anticipating the next nutrient to the quantitative decision of how much of the relevant enzymes to induce. More work is needed to link abstract computations to these lower mechanistic levels.

To represent the structure of meta-changing environments, our model posited a finite number of hidden states that drive nutrient fluctuations. The number of hidden states was fixed in advance, but nonparametric dynamic Bayesian models offer a principled alternative (*Fox et al., 2011*; *Johnson & Willsky, 2013*). In these models, the number of hidden states is learned from observation. Recent work in computational linguistics (*Borschinger & Johnson, 2011*) proposed a particle inference algorithm for a nonparametric dynamic Bayesian model of word segmentation, a task that, like nutrient adaptation, has to be performed in real-time. It would be interesting to investigate whether molecular kinetics can implement such nonparametric Bayesian inference procedures.

While we have focused on glucose-galactose environments, our framework generally applies to environments that change too quickly for mutation and natural selection to take hold. This is distinct from cases where natural selection (e.g., through experimental evolution, as in *Mitchell et al., 2009* and *Tagkopoulos, Liu & Tavazoie, 2008*) rewires circuits genetically to better respond to the predictable structure of the environment. It remains open how inference-based adaptive strategies that apply on short timescales can be implemented at the molecular level, either in natural or engineered cellular circuits. The molecular mechanisms needed to implement these strategies are likely to be epigenetic, based in chromatin (*Stockwell, Landry & Rifkin, 2015*) or stable protein inheritance (*Jarosz et al., 2014*).

We have proposed a design for one critical part of an adaptive inference circuit, which can be supported by a variety of molecular mechanisms. Our circuit design can be implemented using transcriptional, post-transcriptional or epigenetic chromatin-based regulation. The choice of mechanism will determine the timescale and precision of the adaptive response. More work is needed to understand the precision and reliability of the circuit we proposed in the presence of gene expression variability and cell division. A computational account of circuits that can track the state needed for probabilistic inference may also apply to neuronal circuits.

Recent work argued compellingly for developing methods that "compile" abstract computational problems, like probabilistic inference, to molecular descriptions that are physically implementable (*Napp & Adams, 2013*). In this work, a scheme was proposed for solving exact inference for probabilistic graphical models using chemical reaction networks, with DNA strand displacement as the physical instantiation (*Napp & Adams, 2013*). This choice of substrate is implausible as a mechanism for cellular computation,

however (and striving for exact as opposed to approximate inference may be too restrictive for many computational problems). In a different approach, an intracellular kinetic scheme that implements a real-time probabilistic decision procedure for a simple continuously changing environment was proposed (*Kobayashi, 2010*). An open challenge is to extend these schemes to handle structured environments, such as the meta-changing environments we have considered, and to define the molecular components would be needed to build these circuits *in vivo*.

Real-time inference algorithms, such as particle filtering, have the potential to guide the construction of synthetic cellular circuits that adapt to rich changing environments. Since particle filtering algorithms rely on noise, these procedures point to areas where biochemical noise (in gene expression or protein interactions) would not only be tolerated, but would in fact be required for inference to work. These algorithmic features may inform the design of synthetic circuits that implement probabilistic computation out of noisy molecular parts.

## ACKNOWLEDGEMENTS

We thank Matt Johnson, Tommi Jaakkola, Bo Hua, Jenny Chen, Nikolai Slavov, Andrew Bolton, Eric Jonas, Lauren Surface, Ariella Azoulay and Josh Tenenbaum for helpful discussions. We thank Jue Wang for sharing growth rate measurements from 61 yeast strains.

### Funding

The authors received no funding for this work.

### Competing Interests

The authors declare there are no competing interests.

### Author Contributions

- Yarden Katz conceived and designed the experiments, performed the experiments, analyzed the data, wrote the paper, prepared figures and/or tables, reviewed drafts of the paper, conceived project.
- Michael Springer conceived and designed the experiments, analyzed the data, reviewed drafts of the paper.

### Data Availability

Code and rawdata: https://github.com/yarden/paper_metachange.

### Supplemental Information

Supplemental information for this article can be found online at http://dx.doi.org/10.7717/peerj.2716#supplemental-information.

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
