# Peer review of "Probabilistic adaptation in changing microbial environments"

_PeerJ, doi:10.7717/peerj.2716_

## Round 0.1 · original submission · Minor Revisions

Please address the reviewer comments, with special attention to the comment by Reviewer 1 on the validity of the findings. In addition, do clarify to the reader the overall contribution of this work.

·

Basic reporting

Please make sure that the reference format conforms to PeerJ's requirements.

Minor comment about Fig. 1: It is unclear in which way the three lower panels of Figure 1A (temperature, pH and pO2) are related to the top panel. Having them in the same panel makes them appear as though they are correlated. Consider revising Fig. 1A.

Minor comment about Fig. 2/page 7: It should be mentioned somewhere how the values of the growth rate V11 and V22 in R($\pi_1$), R($\pi_2$), and R($\pi_3$) are related to the $\mu_{glu}$ and $\mu_{gal}$ values presented in the main text and Fig. S1.

Experimental design

Showing the population size vs. time in Figs. 4B, 5B, and S3B is not very instructive. Consider presenting them as plots of the average growth rate vs. the p1/p2 parameters instead. For instance, showing how the dependence of each strategy's growth rate on p1/p2 for fixed p2/p1 would tell us about how beneficial each strategy is as the underlying meta-changing environment is evolving.

This could help better quantify how much an organism may benefit from using a posterior predictive strategy, and would add more depth to some of the paper's claims --eg. on line 410: "We found that across different settings of p1 and p2, the posterior predictive policy *generally* results in *substantially larger* population sizes compared with other policies (Figure 4B)" or line 449: "In all cases, the posterior predictive policy produced *larger* population sizes than other policies".

Validity of the findings

The section on "Molecular circuit design for inference-based adaptation" is highly speculative and does not provide much insight. Although the molecular circuit put forward is mathematically correct and its analysis is sound, using this strategy seems very inefficient and is unlikely to be used by an organism: sustained production of a sensor molecule for a metabolite that is not present can become a huge metabolic burden as the number of metabolites is increased beyond 2 or 3, especially considering that yeast can grow on a much larger number of substrates than that. Please consider revising this section.

Additional comments

This is a clear, well-written paper. Besides my few comments above, this article is scientifically sound and the conclusions are well-supported by data. This article should be published in PeerJ.

Reviewer 2 ·

Basic reporting

No comments

Experimental design

No comments

Validity of the findings

No comments

Additional comments

In this manuscript, the authors propose that the problem of adaptive growth in certain structured changing environments can be viewed as probabilistic inference. The authors develop a model and use it to perform simulations. The manuscript is well written, the figures look nice, the calculations are easy to follow, and the methodology seems to be correct. The manuscript, however, is written in such a way that it is not obvious to discern its main contribution. This is particularly important because there are many published articles that have already proposed similar ideas and used similar models. What is the main result of the manuscript? Can it be expressed in a couple of sentences in the abstract? Is it the application of the particle filtering algorithm to the cell growth dynamics? What is new with respect to previous work?

---

## Round 0.2 · accepted · Accept

Thank you again for addressing the reviewers' concerns.